# NPM 1 Mutations in AML—The Landscape in 2023

**DOI:** 10.3390/cancers15041177

**Published:** 2023-02-12

**Authors:** Naman Sharma, Jane L. Liesveld

**Affiliations:** 1Department of Hematology and Oncology, University of Massachusetts-Baystate Medical Center, Springfield, MA 01199, USA; 2Department of Medicine, Hematology/Oncology, James P. Wilmot Cancer Institute, University of Rochester, Rochester, NY 14642, USA

**Keywords:** nucleophosmin (NPM1) mutation, acute myeloid leukemia (AML), menin inhibitors, exportin 1, XPO1 inhibitors, minimal residual disease (MRD), CAR-T cells, arsenic trioxide

## Abstract

**Simple Summary:**

The nucleophosmin 1 (NPM1) gene is mutated in approximately one-third of newly diagnosed acute myeloid leukemia (AML) cases. NPM1mutated AML (NPM1*mut*-AML) has been reclassified as a distinct entity in the 2022 World Health Organization (WHO 2022) and European Leukemia Network (ELN 2022) classification on myeloid neoplasm requiring >10% of leukemic blasts for diagnosis. Clinically, it presents with high white cell counts, blast percentage and frequent extramedullary involvement. In the current era of precision medicine, it carries important diagnostic, clinical and prognostic implications. In this review, we aim to summarize the pathophysiology of the NPM1*mut* in AML, diagnostic and prognostic challenges and treatment strategies including the use of novel targeted therapeutic agents.

**Abstract:**

Acute myeloid leukemia (AML) represents 80% of acute leukemia in adults and is characterized by clonal expansion of hematopoietic stem cells secondary to genomic mutations, rendering a selective growth advantage to the mutant clones. NPM1*mut* is found in around 30% of AML and clinically presents with leukocytosis, high blast percentage and extramedullary involvement. Considered as a “gate-keeper” mutation, NPM1*mut* appears to be a “first hit” in the process of leukemogenesis and development of overt leukemia. Commonly associated with other mutations (e.g., FLT 3, DNMT3A, TET2, SF3B1), NPM1 mutation in AML has an important role in diagnosis, prognosis, treatment and post-treatment monitoring. Several novel therapies targeting NPM1 are being developed in various clinical phases with demonstration of efficacy. In this review, we summarize the pathophysiology of the NPM1 gene mutation in AML, clinical implications and the novel targeted therapies to date.

## 1. Introduction

Acute myeloid leukemia (AML) is characterized by the clonal expansion of hematopoietic stem cells (HSCs) and progenitor cells secondary to genomic aberrations, resulting in selective growth advantage, thereby impeding normal hematopoiesis. AML represents about 80% of acute leukemia cases in adults [1]. In the current age of precision medicine, AML is characterized based on various cytogenetic and molecular abnormalities, as these have important prognostic and treatment implications [2]. Nucleophosmin 1 (NPM1) is one of the commonly mutated genes and is found in approximately 30% of adult leukemia cases [3]. Due to its unique features, NPM1*mut*-AML is recognized as a distinct entity based on this defining genetic abnormality, irrespective of blast counts per the 2022 World Health Organization (WHO) classification on myeloid neoplasms [4]. In this review, we focus on the structure and pathophysiology of the NPM1*mut* in AML and on clinical implications including novel treatment approaches in AML with NPM1 mutation.

## 2. NPM1 Structure and Functions

NPM1 is a multifunctional nuclear chaperone protein shuttling between the nucleus and cytoplasm with primary nucleolar localization. Structurally, NPM1 wild type (NPM*wt*) is a 37 kDa protein with a hydrophobic N-terminus, a central acidic core and basic C-terminus. The N-terminus contains two leucine rich nuclear export signals (NES). The central acidic core (contains aspartic and glutamic acid) is responsible for histone binding and ribonuclease activity and has a bipartite nuclear localization signal (NLS), responsible for nuclear localization of the NPM1*wt*. The C-terminus contains a basic region involved in binding to nucleic acid. The region also contains two tryptophan residues (W288 and W290) responsible for the nucleolar localization signal (NoLS) and interaction with ribosomal DNA [5,6,7,8]. In normal physiological conditions, C-terminus NoLs dominate and direct the nucleolar location of the NPM1*wt*. The NPM1*wt* participates in ribosomal biosynthesis and cell growth by mediating nuclear export and incorporation of 5 S ribosomal RNA into the 60 S ribosomal subunit [9]. NPM1*wt* also binds to the centrosome and prevents uncontrolled duplication during the resting phase of the cell cycle. It briefly dissociates and permits limited centrosome duplication during S and G2 phases of the cell-cycle and re-binds again resulting in each daughter cell containing a single NPM1-bound centrosome post mitosis [10]. In case of oxidative injury, NPM1 via its interaction with p53 increases cellular stability and promotes growth arrest thereby preventing mutagenesis [11]. In summary, NPM1*wt* is a nucleolar protein essential for various phases of the cell cycle and plays an essential role in maintaining cellular homeostasis [12].

## 3. NPM1 Mutation and Clinical Implications

Mutations in NPM1 are always heterozygous and result from 4-bp insertions causing a frameshift mutation at the C-terminus resulting in loss of tryptophan (W288 and W290 or W290 alone) and a gain of new NES leading to disruption of the folded helix structure with loss of NoLS, which shifts the balance towards nuclear export and cytoplasmic localization of NPM1*mut* [12]. This extra-nuclear transfer is mediated by interaction between mutated NPM1 and exportin 1 (XPO1). Exportin 1 (previously Crm1p), a nuclear transporter first identified in yeast (S. cerevisiae) with homologous functions in humans, acts as a direct carrier mediating the export of NES containing protein into the cytoplasm [13]. NPM1*mut*-AML are also associated with a high expression of the homeobox (HOX) gene and its cofactors MEIS1 and PBX3 [14], resulting in increased self-renewal of leukemic clones [15]. Novel agents targeting XPO1 and HOX, which affect the intranuclear relocalization of the mutated NPM1 are therapeutic agents of increasing interest and discussed later in this review (Figure 1).

Almost all mutations in NPM1 are restricted to exon 12 [3], but rarely (<1%) can involve exon 6, exon 9 [16] and exon 11 [17], also leading to cytoplasmic delocalization of NPM1*mut*. The NPM1*mut* can be detected using molecular tests (next generation sequencing—NGS) and by immunohistochemistry (IHC). It is important to acknowledge that the commercially available molecular panels (NGS) are designed to detect the exon 12 mutation only, and therefore emphasis should be maintained to rule out any discrepancy between IHC and molecular studies to avoid wrong assignment to an ELN category. The aberrant cytoplasmic dislocation of NPM1*mut* results in derangement at the cellular level with uncontrolled centrosome duplication, inhibition of tumor suppressor genes, proteolytic activities of caspase 6 and 8, defective DNA repair and activation of the *Myc* oncogene [18], promoting myeloproliferation and leukemogenesis (Figure 1). Animal studies have shown this event to be insufficient to result in full blown disease in all cases, suggesting the need for a second mutation for progression to overt leukemia [19,20]. Therefore, given the pleomorphic function of NPM1*wt*, the aberrant cytoplasmic dislocation of the protein can be considered as the critical “initial” step in the process of leukemogenesis.

NPM1*mut*-AML at diagnosis presents with high blast percentages, elevated white cell and platelet counts, increased extramedullary involvement, and is commonly associated with a normal karyotype [21]. A minority of patients may have additional minor secondary chromosomal abnormalities, the most frequent being +8, +4, del 9q and +21 and abnormal karyotype in around 15%, without any prognostic implication in terms of overall survival (OS) [22]. NPM1*mut* is restricted to myeloid cells and not found in the B and T cells in the peripheral blood or bone marrow [23]. The cytoplasmic NPM1 is specific to AML [3]. Although NPM1*mut* have been described in chronic myelomonocytic leukemia (CMML), these cases all progressed to AML in a short interval after diagnosis. Unlike the mutations associated with clonal hematopoiesis (e.g., DNMT3A, TET2, SF3B1 etc.), NMP1*mut* acts as a “gate-keeper” mutation contributing to leukemogenesis, and acquisition of these mutations appears to be a “first hit” in the development of overt leukemia [24].

### 3.1. NPM1mut and Other Associated Mutations

*3.1.a—NPM1mut*-AML without any other genetic abnormality is considered a favorable category [2] with improved disease-free survival (DFS), OS, and lower relapses, except in older populations >65 years of age [25].

*3.1.b—FMS-like tyrosine kinase 3 (FLT3*) is a tyrosine kinase receptor responsible for hematopoietic cell proliferation and survival on activation. It has two different types of mutations: internal tandem duplication of the juxta membrane (FLT3-ITD) and point mutation or deletion of tyrosine kinase domain (FLT3-TKD); these mutations are common in *NPM1mut*-AML compared to wild type and are now reclassified as an intermediate risk category per the European Leukemia Net (ELN) 2022 classification, regardless of the allelic ratio (change from ELN 2017, where NPM1*mut* with low ITD < 0.5 was considered as favorable risk). In this co-mutation setting, FLT3 mutation is considered as a secondary mutation aiding in leukemogenesis. In these cases, since diverse FLT3-ITD mutants were identified against one NPM1 mutant, this indicates the evolution of various FLT3 mutants evolving from a single NMP1 mutated clone [21,26].

*3.1.c—DNMT3A* mutation associated with clonal hematopoiesis is found in ~60% of NMP1*mut*-AML [27]. DNMT3A, found increasingly in aging populations without overt leukemia [28], is a driver gene which precedes NMP1*mut*, providing a self-renewal advantage to the leukemic clones. DNMT3A as a clonal hematopoiesis of undetermined potential (CHIP) mutation is not considered a high-risk mutation per the ELN 2022 leukemia classification, and it responds optimally to standard therapy with long term complete remission and cure [29].

*3.1.d—Mutation in NPM1, FLT3 and DNMT3A* AML with three-way interaction among NPM1, DNMT3A and FLT3 occurs in around 6% of AML and is characterized by high leukemia stem cell (LSC) frequency, aberrant immunophenotype (with low CD34 and high CD56) and overexpression of hepatic leukemia factor (HLF) [30]. HLF is a novel leukemia stem cell regulator enhancing the self-renewal of leukemic stem cells by upregulating *HES1* (Hairy and Enhancer of Split 1-a NOTCH target) and cell cycle inhibitor p57, thereby maintaining the leukemia stem cells in the quiescent phase and preventing them from entering the cell cycle. This triple mutated genotype results in aggressive chemo-resistant disease due to impaired nucleosome remodeling and overall poor prognosis with high relapse rate, lower disease-free survival, and overall survival [31,32].

*3.1.e—NRAS* mutation is found in ~20% of NPM1*mut*-AML and precedes NPM1 mutation, with preferential association with NRAS^G12/13^ (not NRAS^Q61^) and is associated with granulocytic differentiation and favorable outcomes when co-mutated with DNMT3A [31].

*3.1.f—IDH 1 and 2 mutations* are found in 16% of de novo AML with advanced age, low WBCs, high platelets, and normal cytogenetics. Around 25% of NPM1*mut*-AMLs have an IDH mutation. In a retrospective analysis of 129 patients, the presence of IDH and NPM1 mutations, with un-mutated FLT3 (IDH1^+^/NPM1*mut*^+^/FLT3^−^), was associated with inferior outcomes in terms of relapse free survival and overall survival when compared to wild type IDH (5 years RFS = 37% vs. 67%, OS = 41% vs. 65%) [33].

*3.1.g—TET2* mutation is found in ~16 % of de novo AML [31], and it presents with high white blood count and blast percentage along with normal karyotype and is mutually exclusive with IDH mutation. In NPM1*mut*-AML, 20% of patients will have TET2 mutation and this carries poor outcome in high-risk cytogenetics, or with NPM1*mut* with FLT3-ITD and isolated TET2 mutation with NPM1*wt* (i.e., TET2^+^/ NPM1^−^) [34].

*3.1.h—NPM1mut-AML with multi-lineage dysplasia* (defined as dysplasia in more than or equal two cell lines) is encountered in ~23% cases of NPM1*mut*-AML [35]. It is important to distinguish this from AML-myelodysplasia related (AML-MR; previously called AML with myelodysplasia related changes) due to implications on treatment induction and post-remission consolidation [36]. This distinction is based on the history of exposure to cytotoxic therapies, dysplastic marrow and detection of specific cytogenetic and molecular abnormalities associated with myelodysplasia (e.g., complex karyotype, 5 q deletion, 7q deletion, monosomy 7, ASXL1 mutation, SF3B1 mutation, etc.) [4]. In absence of any myelodysplastic associated features these cases are diagnosed as NPM1*mut*-AML and treated accordingly as the presence of multilineage dysplasia has no impact on prognosis and outcomes [37].

*3.1.i—NPM1mut-AML with abnormal karyotype* Abnormal karyotype is found in ~17% cases and represents secondary events in the disease evolution [38], impacting prognosis and outcomes. In a large retrospective pooled analysis (n = 2426) of NPM1*mut*-AML with low or absent FLT3-ITD, patients with adverse cytogenetics (per ENL 2017) had worse outcomes with high relapse rates, low EFS and OS, thereby offsetting the favorable outcomes of NPM1*mut*-AML with wild type FLT3. The poor outcomes were observed despite consolidation treatment with allo-transplant at CR1 (compared to consolidative chemotherapy) with similar survival rates (66% vs. 67%); a benefit was observed only in lowering the incidence of relapse (23% vs. 41%, *p* < 0.05), but with higher non-relapse related mortality (18% vs. 7.5%). Subgroup analysis looking into specific cytogenetic aberration associated with poor outcomes was statistically inconclusive (owing to small sample size), but monosomies and deletions 5 and 7 were noted to have poor outcomes [38]. However, the presence of these cytogenetic abnormalities will fulfill criteria for AML-MR and would be considered high risk [4].

*3.1.j—NPM1mut-AML with BCR:ABL1* this rare entity is reported in around 0.5–1% of cases [39,40] and is classified as AML with defining genetic abnormalities per WHO classification. This requires >20% blast for diagnosis to avoid overlap with chronic myeloid leukemia (CML) [4]. It is challenging and critical to distinguish this entity from CML with blast crisis (presenting with >20% blasts). In these cases, careful assessment of clinical history (presence or absence of basophilia, splenomegaly etc.) and molecular analysis (transcript p190 and associated NPM1 mutation found in AML) is useful in guiding the diagnosis. When both mutations are present, the NPM1 mutation supersedes the BCR: ABL1 and is considered as a favorable risk category (ELN 2022 does not specifically comments on this category, in contrast to AML with BCR: ABL1 with NPM1*wt* which is considered as adverse risk). These patients are managed as *de-novo* NPM1*mut*-AML rather than as AML with BCR: ABL1; although high quality randomized studies are needed to confirm this practice. The role of tyrosine kinase inhibitors in this entity remains to be defined [41].

To summarize, multiple mutations often co-exist and therefore it is necessary to consider the molecular interactions for accurate risk stratification. Focusing on this unmet need, Sanchez and colleagues developed a machine learning model *(HARMONY Alliance Platform*) aiming to better risk stratify NPM1*mut*-AML [42]. In their analysis of 1093 patients, around 85% of patients carried at least three gene mutations (in decreasing order: DNMT3A, FLT3-ITD, NRAS, TET2). Based on their analysis, the survival was directly dependent on the type of molecular profile e.g., triple mutated (NPM1*mut*+ FLT3-ITD+ *mut*DNMT3A) have the worse outcome (2-year OS-33%) similar to TP53 mutated AML. Interestingly, not all FLT3 mutations conferred bad outcomes as when co-existing with mutated IDH and wild type DNMT3A (FLT3-ITD+ *mut*IDH+ *wt* DNMT3A), excellent outcomes were observed with two-year survival reaching around 80%. Although still awaiting clinical validation, these findings will potentially aid clinical decision making as to whether to proceed to stem cell transplantation on first remission. Other genetic alterations that rarely co-exist with NPM1 mutation are partial tandem duplication in the mixed lineage leukemia (MLL-PTD) gene [21], and mutations in RUNX1, CEBPA and TP53 [43].

### 3.2. NMP1mut AML and Treatment Implications

Clinically, AML with NPM1 mutation often presents with a high bone marrow blast percentage. The blasts often have reduced CD34 and increased CD33 expression [44]. Fit adults (≤60 years, transplant eligible) are treated with intensive chemotherapy, often an anthracycline and cytarabine followed by consolidation with or without a stem cell transplant. In NPM1*mut*-AML without other mutations, many clinicians do not recommend transplantation in the first CR since this is considered a favorable prognosis leukemia [45]. As CD33 is overexpressed in NPM1*mut*-AML, anti-CD33-gemtuzumab-ozogamicin (GO) in young fit adults (<60 years) has been tried with mixed results. In one randomized control trial (ALFA-0701), the use of GO in *de-novo* AML during induction and two additional doses in consolidation resulted in improved relapsed free and overall survival without an increase in toxicity. In this study, only one-third of patients had NPM1 mutation [46]. In another phase 3 trial with NPM1*mut*-AML (AMLSG 09-09), which enrolled about 300 patients in each arm (all NPM1*mut*-AML), patients were randomized to induction chemotherapy (idarubicin, cytarabine, etoposide and all-trans retinoic acid), with or without GO (day 1 of induction and consolidation). The intention to treat analysis did not meet its primary endpoint of improvement in EFS (HR 0.83, CI 0.56–1.04, *p* = 0.1). Subgroup analysis showed EFS benefit in unmutated FLT3 cases [47]. Exploratory analysis of this trial showed a benefit with GO in lowering the incidence of cumulative relapse rate (at 4 years relapse rate 30% vs. 46%; HR 0.66; CI 0.49–0.88; *p* < 0.5) that was attributed to better clearance of the NPM1 mutated transcript and higher MRD (measurable residual disease) negative rates in the GO arm (56% vs. 41%) [48]. Based on these findings, per National Comprehensive Cancer Network (NCCN) guidelines, GO is an option in favorable risk AML (category 2B recommendation).

NPM1*mut*-AML with an FLT3 mutation should receive midostaurin in addition to standard chemotherapy followed by consolidation with allo-HSCT per the RATIFY trial*,* especially with high FLT3-ITD allelic ratio (>0.5), as studies have shown that allo-transplant at first complete remission (CR1) improved both relapse-free and overall survival irrespective of the FLT3 allelic ratio [49,50].

Historically, older fit patients (>60 years) with NPM1*mut*-AML treated with intensive chemotherapy followed by reduced intensity conditioning and non-myeloablative allo-HSCT experience 15–20% long term survival [51,52,53]. In older unfit patients with NPM1*mut*-AML, combining venetoclax with a hypomethylating agent (HMA) induced composite complete remission (i.e., complete response and complete response with incomplete hematological recovery) in approximately 66% of cases and has emerged as the new standard of care in this setting [54]. At a median follow-up of 43 months, sustained benefit has been seen with median OS at 14.7 months (with azacitidine and venetoclax) versus 9.6 months (with azacitidine alone). The benefit was even more pronounced with measurable residual disease (MRD) <10^−3^, with median OS at 34 months versus 18.7 months based on whether an MRD level of <10^−3^ was achieved [55].

In the case of relapsed AML, the prognostic impact of persistent NPM1*mut* is unclear. In a recent retrospective study of 1722 relapsed AML patients (12% with NPM1*mut*), unlike the case in *de-novo* AML, NPM1*mut* had no impact on RFS or OS irrespective of salvage therapy utilized. However, the addition of venetoclax to salvage regimens resulted in a significant and clinically meaningful benefit in RFS and OS (median RFS: 16 vs. 5 months, median OS: 14.7 vs. 6 months). The benefit was further improved when a consolidative allo-transplant was performed (median RFS with and without HSCT 20.7 months vs. 4 months; and median OS 22 vs. 8 months) [56].

### 3.3. Upcoming Novel Therapies on Horizon

#### 3.3.1. Exportin Inhibitors

Exportin 1 (XPO1) is involved in nuclear export and cytoplasmic localization of mutated NPM1 resulting in disruption of cellular homeostasis and increased proliferation and survival of leukemic clones. Inhibition of XPO1 (Figure 1) is being explored as a therapeutic strategy and a new class of drugs called selective inhibitors of nuclear export (SINE) are undergoing therapeutic trials. Selinexor, first in this class, was examined in a phase 1 study, preventing nuclear export of NMP1 mutants. This study showed low efficacy that was attributed to dose limiting toxicities (grade ≥ 3 fatigue, anorexia, neutropenia, and thrombocytopenia) [57]. A second generation XPO1 inhibitor (SINE), eltanexor (KPT-8602), with an advantage of lower CNS penetration (thereby offsetting the side effects of anorexia) permitting frequent dosing and higher drug concentration, resulting in stable inhibition of NPM1 and XPO1 interaction, has shown promise in preclinical studies [58]. The efficacy of SINE is further enhanced when combined with BCL2 inhibitors as seen in preclinical models [59] and is currently being evaluated in early phase trials (NCT03955783).

#### 3.3.2. Menin-KMT2A Inhibitors

NPM1*mut*-AMLs have an overexpression of the HOX gene (HOXA9) and its cofactor MEIS1, leading to leukemogenesis via CEBPα (CCAAT enhancer-binding protein alpha) and lysine methyl transferase 2 (KMT2A) gene (previously known as mixed lineage leukemia 1 gene—MLL1) [60,61]. KMT2A, a transcriptional regulator which binds to menin in the N-terminal and the Menin–KMT2A complex, then regulates HOX9 [62]. A recent study exhibited that the expression of HOX is directly dependent on NPM1 mutants [15], making menin-KMT2A inhibition a potential therapeutic option in NPM1*mut*-AML (Figure 1), with efficacy seen in animal studies [63]. A recently published early phase (1/2) trial with revumenib (SNDX-516) in heavily pretreated patients (four previous lines of therapy, 46% having undergone transplant) had an overall response rate (ORR) that was encouraging, at 60% in KMT2A or NPM1*mut*-AML [64]. Similar results were seen with another menin inhibitor (ziftomenib: KO 539) with ORR reaching ~40% [65], with common side effects being cytopenia, infections and diarrhea.

#### 3.3.3. PD-1 Inhibition

NPM1*mut* serves as a driver mutation caused by a 4 bp frameshift insertion at the C-terminus and results in a longer mutated protein which acts as a neo-antigen. This is expressed in the malignant phenotype and is less likely to be lost due to immune editing, thereby making this protein a potentially ideal target for immunotherapy [66]. High PD-L1 expression has been found in NPM1*mut*-AML (mainly in the leukemic progenitor cells; CD34+, CD38−) [67], with activity seen with nivolumab in in vitro studies [68]. The action of HMAs against the leukemic cells is mediated by upregulation of tumor antigens expressed by the class 1 major histocompatibility complex and co-stimulatory molecule, resulting in the destruction of cancer cells, but eventual upregulation of program cell death protein (PD-1) and cytotoxic T- lymphocyte associated protein 4 (CTLA-4) leads to immune escape and drug resistance [69]. To overcome this resistance mechanism, the role of pembrolizumab (a PD-1 inhibitor) is currently being evaluated in combination with azacitidine in molecular relapse in an ongoing phase 2 trial. It is not known if this would have preferential response in NPM1 mutated cases.

#### 3.3.4. CAR-T/T-cell Receptor (TCR) Therapy

Neo-antigens formed due to NMP1 mutation are uniformly expressed and have been targeted using chimeric antigenic receptor constructs. Multiple peptidomes (e.g., CLAVEEVSL, AIQDLCLAV) have been identified when translated in an alternating reading frame [66,70] and these neo-peptides are presented by human leucocyte antigen-A2 (HLA-A2). Chimeric antigen receptor constructs selectively identifying the NPM1*mut*–HLA–A2 complex have shown strong activity in preclinical studies [71].

CD123 and CD33 are also considered ideal targets for treatment as they are strongly expressed by both NPM1*mut*-AML blasts and are preserved at disease relapse. They are also co-expressed on the hematopoietic stem cells (CD33) and vascular endothelial cells (CD133). CAR constructs targeting these resulted in severe prolonged myeloablation and capillary leakage, limiting the clinical use. To mitigate this toxicity, modified CAR constructs with controlled selective activation against CD123/CD33 positive leukemic cells have shown efficacy in preclinical studies [72].

At present, challenges with the lack of leukemia specific cell surface antigen in myeloid malignancies, resulting in significant off-target toxicities and an inhibition of T cell expansion by leukemic blasts limits the clinical use of cellular adoptive therapies in AML to early phase clinical trials [73] (NCT05252572, NCT 05239689).

#### 3.3.5. Arsenic Trioxide (ATO) and All-Trans Retinoic Acid (ATRA)

Degradation of oncoprotein with ATO and ATRA resulted in high cure rates in acute promyelocytic leukemia. NPM1*mut* results in cytoplasmic localization of mutated protein with high sensitivity to oxidative stress and apoptosis due to substitution of W-288 by cysteine at the C-terminus [74]. When treated in combination with ATO and ATRA, increased apoptosis is seen in the leukemic cell lines via proteasome mediated degradation, p53 activation [75]. In terms of treatment implication, the subgroup analysis of the AML HD98B trial (23% cases with NPM1*mut*) showed that the addition of ATRA to chemotherapy improved outcomes in RFS and OS (HR = 0.27. and 0.28 respectively) in elderly patients with mutated NPM and wild FLT3 [76], with the benefit attributed to the dosing and the sensitizing effect of ATRA on chemotherapy. Table 1 lists the ongoing advanced phase trials in NPM1*mut*-AML.

### 3.4. Post Treatment MRD Assessment

NPM1*mut* is frequent, not associated with clonal hematopoiesis, eradicated with treatment, and re-appearance precedes hematological relapse, thereby making an ideal target for MRD monitoring [77]. Patients achieving complete remission by morphological assessment alone may still have a low level of leukemia burden that can predict long-term outcomes [77]. Several studies, both in children and adults with AML, have demonstrated a correlation between MRD positivity and the risk of relapse, as well as the prognostic significance of MRD measurements at end of the induction therapy [78,79]. The timing and frequency of MRD detection varies based on the type of leukemia and the protocol used but is usually performed at end of initial induction and consolidation. Per ELN 2022, in NPM1*mut*-AML, MRD assessment is performed after two cycles of chemotherapy and end of treatment [2]. The most frequently employed method for MRD assessment includes real-time polymerase chain reaction (RQ-PCR) assay and multicolor flow cytometry (MFC) specifically designed to detect abnormal MRD immunophenotypes. The threshold to define MRD+ and MRD- states depends on the technique and subgroups of AML (for RQ-PCR MRD+, this is defined as failure to reduce mutant transcript levels by >3 log reduction in bone marrow, or >4 log reduction in peripheral blood) [2]. NGS based assays are not routinely used at present, due to overall superior sensitivity of the PCR-based assay and flow cytometry.

Patients who are MRD+ at the end of consolidation benefit from allo-HSCT in terms of relapse rates and overall survival [80]. MRD assessment is also recommended before the allo-HSCT, as persistent positivity is associated with poor outcomes with increased relapse and low survival independent of cytogenetics [81]; however, at this time, there is no evidence supporting additional courses of treatment before transplant to achieve an MRD negative state. For allo-transplant, myeloablative conditioning (MAC) is preferred over reduced intensity conditioning (RIC), as the former has better outcomes in terms of relapse and overall survival (3-year OS 61% vs. 43%) [82]. As the NPM1 transcripts in bone marrow or peripheral blood can reliably predict hematological relapse (usually within 3 months), ELN recommends patients undergoing standard chemotherapy to be tested every 3 months from bone marrow (every 4–6 week if peripheral blood) [2] during the first 2 years of follow-up. Even though molecular relapse predicts hematological relapse, molecular MRD may remain detectable at low levels without clinical significance if the values are below the threshold linked to prognosis [83]. New treatment approaches incorporating newer agents and pre-emptive treatment based on molecular relapse are currently being evaluated in various phase 2 and phase 3 trials, as listed in Table 1. Around 5–10% of NPM1*mut*-AML can relapse without detectable NPM1 mutation at the time of relapse [84]. In these cases, the timing of relapse is of importance as relapse occurring years after the initial diagnosis is due to return to the state of persistent clonal hematopoiesis post eradication of the original NPM1 mutated clone, with predisposition to a second hematological neoplasm. On the contrary, if relapse occurs early without detectable NPM1 clones, this could represent t-MDS/AML [85]. A brief overview of NPM1*mut*-AML treatment strategies is summarized in Figure 2.

## 4. Conclusions

The NPM1 mutation is an AML defining mutation with several clinical and prognostic implications. Since its identification, it has been increasingly utilized in the identification and risk stratification of AML. As most often multiple mutations co-exist and affect outcomes, there is an unmet need for a clinical model incorporating these variables in clinical decision making. The role of molecular disease monitoring is well-established and often precedes a hematological relapse. At present there is no role for pre-emptive treatment for molecular relapse without hematological relapse, and the role of immunotherapy along with azacitidine is being evaluated. Attempts are also underway to identify, and target, molecules involved in the pathogenesis with HOX and menin inhibitors. Attempts are being made to identify and target specific leukemia surface antigens using specific CAR constructs and are currently in various preclinical and clinical stages of development with the overall goal to improve outcomes in this AML subtype.

## Figures and Tables

**Figure 1 cancers-15-01177-f001:**
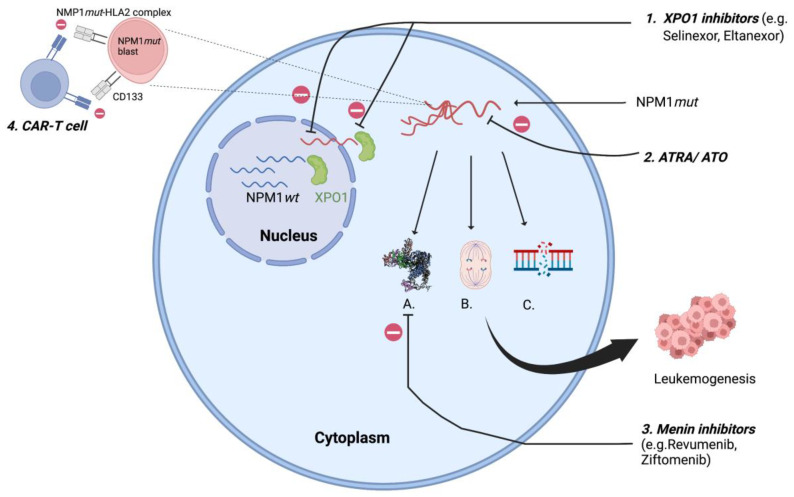
XPO1 transports NPM1*mut* to cytoplasm resulting in A. Homeobox (HOX) overexpression; B. Centrosome duplication; C. defective DNA repair resulting in leukemogenesis; 1. XPO1 inhibitors: inhibits XPO1 mediated cytoplasmic export of NPM1*mut*; 2. ATRA/ ATO: oxidative injury and apoptosis of NPM1*mut* cells; 3. Menin inhibitors: suppresses *HOX* by inhibiting menin- KMT2A complex; 4. CAR-T cell: targets NPM1*mut*-HLA2 complex, CD133. XPO1- exportin 1; ATRA/ATO- all-trans retinoic acid/arsenic trioxide. (Figure created using BioRender).

**Figure 2 cancers-15-01177-f002:**
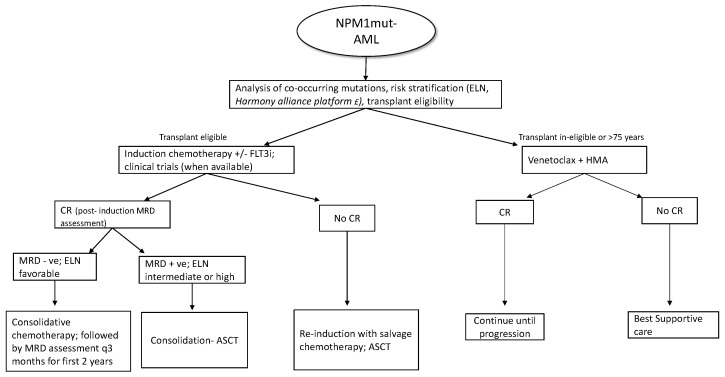
Overview of treatment in NPM1*mut*-AML; (FLT3i- FLT3 inhibitor; CR- complete response; ASCT-allogenic stem cell transplant; HMA- hypomethylating agents, £- pending clinical validation).

**Table 1 cancers-15-01177-t001:** List of ongoing phase 1/2 and 3 trials in NPM1*mut*-AML (£: trials exploring pre-emptive treatment on molecular relapse) (date accessed-January 7, 2023) (ref. Clinicaltrials.gov).

Trial	Intervention	Current Status	Outcomes
1. NCT00893399 (Phase 3)	Chemotherapy with ATRA with or without gemtuzumab- ozogamicin	Completed, final data collection, September 2021	Primary—OS Secondary—EFS, safety, CR post-induction, QoL
2. NCT04689815 ^£^ (Phase 2)	Phase 2, single arm study with oral arsenic trioxide, ascorbic acid and azacytidine for MRD positive for NPM1 AML post consolidation, transplant ineligible	Recruiting	Primary—rate of NPM1 MRD negativity Secondary—DOR, leukemia- free survival (LFS), safety
3. NCT04867928 ^£^ (Phase 2)	Phase 2, single arm, open label- venetoclax and azacitidine on molecular relapse as bridge to allo-HSCT	Recruiting	Primary—efficacy of venetoclax and azacitidine as bridge to transplant
4. *NCT05020665* *(Phase 3)*	Phase 3, randomized, double blind, placebo control oral entospletinib vs. placebo with combination with intensive induction and consolidation chemotherapy in newly diagnosed NMP1 mut-AML	Active, not recruiting	Primary—MRD negative CR Secondary—EFS, RFS, CR, adverse events
5. NCT01237808 (Phase 3)	Low-Dose cytarabine (20 mg/day, SQ BID day 1–7) and etoposide with or without ATRA older patients not eligible for intensive chemotherapy in NPM1*mut*-AML	Completed	Primary—overall survival Secondary—rates of CR, cumulative incidence of relapse, EFS, deaths, adverse events, QoL
6. NCT03769532 ^£^ (Phase 2)	Single arm, pembrolizumab with azacitidine in morphological remission but MRD positive	Recruiting	Primary—EFS Secondary—OS, EFS, safety
7. NCT03031249 (Phase 1,2)	Post induction consolidation with combination of cytarabine, ATO and ATRA vs. cytarabine alone	Unknown	Primary—RFS (up to 5 years from randomization) Secondary—OS, cumulative incidence of relapse
8. NCT04988555 (Phase 1,2)	Open-label, single arm safety and efficacy of DSP-5336 in relapsed/ refractory AML/ALL	Recruiting	Primary—safety Secondary—efficacy
9. NCT04065399 (Phase 1,2)	Open label, single arm study of SNDX-5613 in relapsed/refractory leukemia (AUGMENT-001)	Recruiting	Primary—safety Secondary—CR, median RFS, OS, DOR

(OS—overall survival, DOR—duration of response, EFS—event free survival, MRD—measurable residual disease, CR—complete response, QoL—quality of life, ATRA—all-trans retinoic acid, ATO—arsenic trioxide, RFS—relapse free survival).

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
