# Peer review of "NPM 1 Mutations in AML—The Landscape in 2023"

_cancers, 2023, doi:10.3390/cancers15041177_

Round 1

Reviewer 1 Report

This is an overall nicely written paper summarizing what we currently know about NPM1 mutations in AML. The presence of NPM1 mutations strongly affect prognosis and outcomes and could pave the way for the application of several novel therapeutics.

I have few comments to the authors:

1- I suggest rewriting the simple summary at the beginning; the sentences are long and could benefit from rewording.

2- There are few repetitions across the manuscript, for example related to the prevalence of NPM1mut in AML.

3- While the review addresses several important aspects pertaining to NPM1 mutations, I consider it to be very concise. Therefore I suggest it expands on some currently hot points of discussion for further enrichment:

    A- Diagnostic methods and pitfalls of NPM1 mutations (a paragraph)

    B- Perspective on targeted therapy in NPM1mut beyond Menin inhibitors, namely the role of arsenic trioxide based on molecular and biological data, as well as CAR and TCR T-cell-based therapies against neoantigens created by the NPM1 mutations (CD33, CD123, etc). 

    C- Figure A could benefit from adding other targeted therapies and their corresponding point(s) of targets (Immunotherapy, ATO, TCR gene engineered, etc.)

    D- Figure B's algorithm is very simplified and does not necessarily take into consideration older patients with NPM1mut AML. I suggest adding a section for older patients or those who are transplant ineligible

Reviewer 2 Report

This is a useful review of NPM1 mutations in AML.  It is well-written and fairly comprehensive.  The authors may have noted an abstract at the 2022 American Society of Hematology annual meeting (Hernandez Sanchez et al) that used machine-learning on the Harmony Alliance Platform and proposed a new genetic stratification model based on co-expression of NPM1mut and other mutations (FLT3-ITD, DNMT3A and IDH).  

Minor comments:

Page 12 in the exportin section, the word similarly is used before the sentence concerning a second generation inhibitor.  I would drop the word "similarly" since it does not seem to have a similar profile. 

In the structure and function section (Page 2) it  provides interesting details of NPM1 biochemistry and function but it does not seem to explain "how" it is responsible for control of cell cycle, centrosome duplication, ribosome synthesis, etc.  

There are a few very minor edits needed (dropped articles, etc).  
